# Consensus Control of Linear Parameter-Varying Multi-Agent Systems with Unknown Inputs

**DOI:** 10.3390/s23115125

**Published:** 2023-05-27

**Authors:** Fanglai Zhu, Chengmin Tan

**Affiliations:** College of Electronics and Information Engineering, Tongji University, Shanghai 201804, China; 2033121@tongji.edu.cn

**Keywords:** consensus, linear parameter-varying system, interval observer, unknown input reconstruction, multi-agent systems

## Abstract

This paper investigates the observer-based consensus control problem for linear parameter-varying (LPV) multi-agent systems (MASs) with unknown inputs. Firstly, an interval observer (IO) is designed to generate the state interval estimation for each agent. Secondly, an algebraic relationship is established between the system state and unknown input (UI). Thirdly, an unknown input observer (UIO) capable of generating estimates of UI and the system state has been developed through the algebraic relations. Finally, a UIO-based distributed control protocol scheme is proposed to realize the consensus of the MASs. In the end, to verify the validity of the proposed method, an example of a numerical simulation is given.

## 1. Introduction

In order to deal with the gain scheduling control system where its dynamic behavior is affected by some real-time measurable scheduling parameters, the control theory of LPV systems was first proposed by Shamma in 1988 [1]. On the one hand, LPV systems are more effective in describing nonlinear systems than traditional ones. Based on LPV systems, gain scheduling controllers with time-varying parameters can be designed to gain better control performances than traditional ones. On the other hand, an LPV system can be considered as a special linear system where the coefficient matrix in its state-space description changes with the variations of the scheduling parameters. The matrix variations of an LPV system depend on time-varying parameters, the borrowing of linear systems techniques and theories. Unknown input observers (UIO) for LPV systems can be found in [2,3,4,5,6]. In [2], a UIO for an LPV system is designed using algebraic matrix operations for simultaneous state and unknown-input reconstruction. Li [3] studies UIO design for LPV systems in the presence of bounded error by integrating the decoupling technique and set-membership method. As a special form of UIO, interval observer (IO) designs for LPV systems can be found in [7,8,9,10,11,12]. Khan [11] proposes a new design method of interval state estimator for an LPV system subject to state disturbances and measurement noises using the observability matrix. In [12], the finite-time control design for an LPV system is studied via the interval method, where an interval observer is constructed using an unknown input observer to avoid co-constraints in the estimation error dynamics.

The problem of consensus in MASs has aroused much attention for decades simply because MASs have many practical applications. The basic idea of the consensus problem of a MAS is designing a distributed control protocol based on the exchange of information among agents in order to fulfill the synchronization of agents’ states. Early work utilized synovial sliding technology to address the consensus issues of low-order systems [13,14]. Later, some researches on MASs of event-triggered consensus control schemes were also considered [15,16,17,18,19,20,21]. In [21], for linear MASs whose control input is heterogeneous sector-restricted nonlinearities, the problem of event-triggered consensus was studied. Some consensus problems are dealt with under DoS attacks [22,23,24]. For the leader–followers MASs, consensus control has been studied [25,26,27,28,29]. In [29], in order to achieve the tracking consensus of robot MASs, an optimal sliding-mode control method based on a projection recurrent neural network was proposed. Moreover, in [30], an encryption–decryption scheme is used to realize the fault-tolerant consensus control goal. However, there are few papers that combine MASs with LPV systems [31,32,33,34,35]. In [32], with a limited number of LMIs, a function controller and observer gains with time-varying parameters can be obtained, and the consensus problem for MASs is then solved.

The present paper focuses on UIO-based consensus problems of LPV MASs with unknown inputs. The main contributions of this paper can be summed up in the following points:(1)Using an interval oberver for each follower agent, an algebraic relationship is established between the state and the UI.(2)By combining an unknown input reconstruction (UIR) with a Luenberger-like state observer, we have designed a UIO that can simultaneously obtain the state and UI estimates.(3)By using the state estimation and the UIR, we design a distributed controller for each follower agent, and under the distributed controller, the asymptotic convergence consensus problem is solved. In this way, a UIO-based asymptotic consensus control protocol scheme is put forward.

The paper is organized in the following way: Some preliminaries and the system description are given in Section 2. The IO and UIO designs are provided in Section 3. The distributed consensus controller design method based on LMI is given in Section 4. A numerical simulation example is shown in Section 5. Some conclusions are drawn in Section 6.

## 2. Preliminaries and System Description

In this section, some notations, definitions, lemmas and assumptions are presented. In addition, the model description of the LPV MASs is given.

### 2.1. Preliminaries

**Notation** **1.**
*For matrices $W=\left[w_{ij}\right]\in R^{n\times m}$ and $M=\left[m_{ij}\right]\in R^{n\times m}$, W⩽M if and only if $w_{ij}⩽m_{ij}$ for i=1,⋯n;j=1,⋯,m. Moreover, define $W^{+}=\left[\max\left(0,w_{ij}\right)\right]\in R^{n\times m}$, $W^{-}=\left[\max\left(0,-w_{ij}\right)\right]\in R^{n\times m}$ and $\left|W\right|=\left[\left|w_{ij}\right|\right]\in R^{n\times m}$. Clearly, $W=W^{+}-W^{-}$ and $\left|W\right|=W^{+}+W^{-}$. Furthermore, if W<M, then $W^{-}⩾M^{-}$ and $W^{+}⩽M^{+}$. F≻0 means that the matrix $F\in R^{m\times m}$ is positive definite. $F\underset{-}{≻}0$ means that the matrix $F\in R^{m\times m\;}$ is positive semi-definite. For any vector of $x\in R^{n}$, $\mathrm{diag}\left(x\right)$ represents the diagonal matrix; the elements of x are its diagonal elements. $1_{N}={\left[\begin{array}{cccc} 1 & 1 & \cdots & 1 \end{array}\right]}^{T}\in R^{N}$.*

*In order to generate its first-order derivative information, for a known signal $\beta \left(t\right)$, the following differentiator [36] is used in the design:*

(1)
$$\left\{\begin{array}{c} {\dot{\xi }}_{1}=v_{1},\;\;v_{1}=-\eta_{1}{\left|\xi_{1}-\beta \right|}^{1/2}\cdot \mathrm{sign}\left(\xi_{1}-\beta \right)+\xi_{2}\\ \xi_{2}=-\eta_{2}\mathrm{sign}\left(\xi_{2}-v_{1}\right) \end{array}\right.$$


*Then, the identical estimate for $\dot{\beta }\left(t\right)$ is $\xi_{2}\left(t\right)$ within a finite time frame.*


**Notation** **2.***This paper considers a group of LPV MASs that consists of a leader agent and* N *follower agents. An undirected weighted graph $G=\left(E,V,A\right)$ is used to describe the flow of information among the* N *follower agents, where $A=\left[a_{ij}\right]\in R^{N\times N}$ is the weighted adjacency matrix, $V=\left\{v_{1},\cdots ,v_{N}\right\}$ is the node set and E⊆V×V is the edge set. If $\left(v_{i},v_{j}\right)\in E$, nodes $v_{i}$ and $v_{j}$ are viewed as neighbors of each other, which means they can exchange information with each other, and accordingly, we set $a_{ij}=a_{ji}=1$ if $\left(v_{i},v_{j}\right)\in E$ and $a_{ij}=a_{ji}=0$ otherwise. Self-edges are excluded, that is, $a_{ii}=0$ for all $i\in \left\{1,\cdots ,N\right\}$. Let $L=\left[l_{ij}\right]\in R^{N\times N}$ be the Laplacian matrix of G with $l_{ij}=-a_{ij}$ when i≠j and $l_{ii}=\sum_{j=1}^{N}a_{ij}$ for all $i\in \left\{1,\cdots ,N\right\}$. Let $B=\mathrm{diag}\left(b_{1},\ldots ,b_{N}\right),b_{i}⩾0,i=1,\ldots ,N$. $b_{i}>0$ means that the* i*th follower can interact with the leader. Denote H=L+B.*

**Definition** **1**([37])**.**
*A Metzler matrix is a square matrix in which all its off-diagonal elements are non-negative. If all the eigenvalues of a square matrix have negative real parts, it is called a stable matrix or Hurwitz matrix.*

**Lemma** **1**([37])**.**
*The matrix $A\in R^{n\times n}$ is supposed to be a Metzler and Hurwitz matrix. Moreover, $d_{x}\left(t\right)\in R^{n}$, $d_{x}\left(t\right)⩾0$ for all t⩾0. Then the dynamic system $\dot{x}\left(t\right)=Ax\left(t\right)+d_{x}\left(t\right)$ has solutions satisfying $x\left(t\right)⩾0$$\left(t⩾0\right)$ provided that the initial state $x\left(0\right)⩾0$.*

**Lemma** **2**([38])**.**
*Suppose $\underset{\underline{\;}}{x}\left(t\right)\in R^{n}$, $x\left(t\right)\in R^{n}$ and $\bar{x}\left(t\right)\in R^{n}$ satisfy $\underset{\underline{\;}}{x}\left(t\right)\leq x\left(t\right)\leq \bar{x}\left(t\right)$, $S\in R^{m\times n}$. We have*
$$S^{+}\underset{\underline{\;}}{x}\left(t\right)-S^{-}\bar{x}\left(t\right)\leq Sx\left(t\right)\leq S^{+}\bar{x}\left(t\right)-S^{-}\underset{\underline{\;}}{x}\left(t\right)$$
*If $S\in R^{m\times n}$ is a variable matrix and $\underset{\underline{\;}}{S}\leq S\leq \bar{S}$ for some $\bar{S},\underset{\underline{\;}}{S}\in R^{m\times n}$, then*

$${\underset{\underline{\;}}{S}}^{+}{\underset{\underline{\;}}{x}}^{+}-{\bar{S}}^{+}{\underset{\underline{\;}}{x}}^{-}-S^{-}{\bar{x}}^{+}+{\bar{S}}^{-}{\bar{x}}^{-}\leq Sx\left(t\right)\leq {\bar{S}}^{+}{\bar{x}}^{+}-S^{+}{\bar{x}}^{-}-{\bar{S}}^{-}{\underset{\underline{\;}}{x}}^{+}+{\underset{\underline{\;}}{S}}^{-}{\underset{\underline{\;}}{x}}^{-}$$



### 2.2. System Description and Assumptions

Consider a MAS with *N* LPV agent followers and an LPV leader. The leader agent is characterised as
(2)$$\left\{\begin{array}{c} {\dot{x}}_{0}\left(t\right)=A\left(\theta \left(t\right)\right)x_{0}\left(t\right)\\ y_{0}\left(t\right)=Cx_{0}\left(t\right) \end{array}\right.$$
in which $x_{0}\left(t\right)\in R^{n}$ and $y_{0}\left(t\right)\in R^{p}$ are the system state and output vectors, respectively. The *i*th follower agent is being modelled as
(3)$$\left\{\begin{array}{c} {\dot{x}}_{i}\left(t\right)=A\left(\theta \left(t\right)\right)x_{i}\left(t\right)+B\left(\theta \left(t\right)\right)\left[u_{i}\left(t\right)+d_{i}\left(t\right)\right]\\ y_{i}\left(t\right)=Cx_{i}\left(t\right) \end{array}\right.$$
where $x_{i}\left(t\right)\in R^{n}$ is the state vector, $y_{i}\left(t\right)\in R^{p}$ is the measurement output vector, $u_{i}\left(t\right)\in R^{m}$ is the control input vector and $d_{i}\left(t\right)\in R^{m}$ is the external disturbance vector. Furthermore, $A\left(\theta \right)=\sum_{i=1}^{S}\rho_{i}\left(\theta \right)A_{i}$, $B\left(\theta \right)=\sum_{i=1}^{S}\rho_{i}\left(\theta \right)B_{i}$, where $A_{i},B_{i},\left(i=0,1,\cdots ,S\right)$ and $C\in R^{p\times n}$ are known constant matrices with appropriate dimensions, and $0⩽\rho_{i}\left(\theta \right)⩽1$ satisfying $\sum_{i=1}^{S}\rho_{i}\left(\theta \right)=1$. In addition, we suppose that matrices *C* and B(θ) severally have a full row as well as column ranks. It is also assumed that θ is a scalar parameter that can be measured online.

**Assumption** **1**([38])**.*** $∥x∥\leq \bar{x}$ and $∥y∥\leq \bar{y}$, the scalars $\bar{x}>0$ and $\bar{y}>0$ are known. There are upper bounds ${\bar{x}}_{i}\left(t\right)$ and lower bounds ${\underset{\underline{\;}}{x}}_{i}\left(t\right)$ for the state variables $x_{i}\left(t\right),i=1,\ldots ,N$ of each agent.*

**Assumption** **2**([38])**.**
*There exists a matrix function $E\left(\theta \right)\in R^{n\times p}$, $F\left(t\right)\in R^{n\times n}$ satisfying $F\left(t\right)=F{\left(t\right)}^{T}≻0$ so that $∥y∥⩽\bar{y}$$\left(t⩾0\right)$:*
$$\begin{array}{c} f_{1}I_{n}\underset{-}{≺}≺F\left(t\right)\underset{-}{≺}f_{2}I_{n},f_{1},f_{2}>0\\ \dot{F}\left(t\right)+N{\left(\theta \right)}^{T}F\left(t\right)+F\left(t\right)N\left(\theta \right)+F{\left(t\right)}^{2}+O\underset{-}{≺}0\\ N\left(\theta \right)=A\left(\theta \right)-A\left(\theta \right)C,O=O^{T}≺0 \end{array}$$

**Assumption** **3**([38])**.**
*Let $N\left(\theta \right)\in \Xi$ for all t⩾0, where*
$$\Xi =\left\{N\left(\theta \right)\in R^{n\times n}:N_{0}-\bar{\Delta N}⩽N\left(\theta \right)⩽N_{0}+\bar{\Delta N}\right\}$$
*for some $N_{0}=N_{0}^{T}\in R^{n\times n}$ and $\bar{\Delta N}\in R_{+}^{n\times n}$. The matrix $N_{0}$ has the same eigenvalues as the Metzler matrix $H=\upsilon I_{n}-\Omega$, for a diagonal matrix $\Omega \in R^{n\times n}$ and a certain constant $\upsilon >n{∥\bar{\Delta N}∥}_{\max}$.*

**Assumption** **4**([39,40])**.**
*The pair $\left(A_{0},C\right)$ is observable. Moreover, for all θ,*
$$\mathrm{rank}\left[\begin{array}{cc} sI_{n}-A\left(\theta \right) & B\left(\theta \right)\\ C & 0 \end{array}\right]=n+q$$
*is true for any s meeting with Res⩾0.*

**Assumption** **5**([39,40])**.**
*For all θ, the following rank condition $\mathrm{rank}\left[CB\left(\theta \right)\right]=\mathrm{rank}\left(B\left(\theta \right)\right)=q$ holds.*
*Under Assumption 5, the Moore–Penrose inverse of $CB\left(\theta \right)$ exists, and it is defined as ${\left(CB\left(\theta \right)\right)}^{†}={\left[{\left(CB\left(\theta \right)\right)}^{T}\left(CB\left(\theta \right)\right)\right]}^{-1}{\left(CB\left(\theta \right)\right)}^{T}$.*


**Lemma** **3.**
*Under Assumption 4, for all θ, the pair*

(4)
$$\left(\left[I_{n}-B\left(\theta \right){\left(CB\left(\theta \right)\right)}^{†}C\right]A\left(\theta \right),C\right)$$

*is detectable.*


**Proof.** Because
$$\begin{array}{c} n+q=\mathrm{rank}\left[\begin{array}{cc} sI_{n}-A\left(\theta \right) & B\left(\theta \right)\\ C & 0 \end{array}\right]\\ =\mathrm{rank}\left\{\left[\begin{array}{cc} sI_{n}-A\left(\theta \right) & B\left(\theta \right)\\ C & 0 \end{array}\right]\left[\begin{array}{cc} I_{n} & 0\\ {\left(CB\left(\theta \right)\right)}^{†}CA\left(\theta \right) & I_{q} \end{array}\right]\right\}\\ =\mathrm{rank}\left[\begin{array}{cc} sI_{n}-\left[I_{n}-B\left(\theta \right){\left(CB\left(\theta \right)\right)}^{†}C\right]A\left(\theta \right) & B\left(\theta \right)\\ C & 0 \end{array}\right] \end{array}$$
holds, therefore, under Assumption 4, for all *θ*, we have
$$\mathrm{rank}\left[\begin{array}{c} sI_{n}-\left[I_{n}-D\left(\theta \right){\left(CD\left(\theta \right)\right)}^{†}C\right]A\left(\theta \right)\\ C \end{array}\right]=n$$
which holds for any s in which Res⩾0, which implies that the pair (4) is detectable. □

**Assumption** **6.**
* $d_{i}\left(t\right)$ is bounded by $\underset{\underline{\;}}{d}{}_{i}\leq d_{i}\left(t\right)\leq \bar{d_{i}}$, in which the lower bound $\underset{\underline{\;}}{d}$ and the upper bound $\bar{d}$ are two vectors of known constants. $x_{i}\left(0\right)$ is bounded by $\underset{\underline{\;}}{x}{}_{i0}\leq x_{i}\left(0\right)\leq {\bar{x}}_{i0}$, where $\underset{\underline{\;}}{x}{}_{0}$ and ${\bar{x}}_{0}$ are two constant known vectors.*


**Definition** **2.**
*The leader–follower MAS (2) and (3) is said to be consensus if $\lim_{t\to \infty }\delta_{i}\left(t\right)=0$ can be reached for $i\in \left\{1,\cdots ,N\right\}$, where $\delta_{i}\left(t\right)=x_{i}\left(t\right)-x_{0}\left(t\right)$ is the consensus variable.*

*The main objective of this paper is to design an IO-based distributed controller for each follower agent so that MAS consensus in the sense of Definition 2 can be accomplished under a distributed control protocol scheme.*


## 3. Unknown Input Observer Design via Interval Observer

In this section, for system (3), an IO is designed to obtain the interval estimations for the state variables and the system output. Afterwards, a UIR approach has been developed via the IO.

### 3.1. Design of Interval Observer

To begin with, choose an appropriate invertible matrix T, make a state transformation $z_{i}\left(t\right)=Tx_{i}\left(t\right)$, and then the system (2) can be equated to
(5)$$\left\{\begin{array}{c} {\dot{z}}_{i}\left(t\right)=TA\left(\theta \right)T^{-1}z_{i}\left(t\right)+TB\left(\theta \right)u_{i}\left(t\right)+TB\left(\theta \right)d_{i}\left(t\right)\\ y_{i}\left(t\right)=CT^{-1}z_{i}\left(t\right) \end{array}\right.$$

Now the following IO has been designed for system (5)
(6)$$\left\{\begin{array}{c} {\dot{\bar{z}}}_{i}\left(t\right)=TA\left(\theta \right)T^{-1}{\bar{z}}_{i}\left(t\right)+{\bar{\phi }}_{i}+TB\left(\theta \right)u_{i}\left(t\right)+TE\left(\theta \right)\left(y_{i}-CT^{-1}{\bar{z}}_{i}\left(t\right)\right)\\ {\dot{\underset{\underline{\;}}{z}}}_{i}\left(t\right)=TA\left(\theta \right)T^{-1}\underset{\underline{\;}}{z}{}_{i}\left(t\right)+\underset{\underline{\;}}{\phi }{}_{i}+TB\left(\theta \right)u_{i}\left(t\right)+TE\left(\theta \right)\left(y_{i}-CT^{-1}\underset{\underline{\;}}{z}{}_{i}\left(t\right)\right) \end{array}\right.$$
where ${\bar{\phi }}_{i}=T^{+}{\bar{\tau }}_{i}-T^{-}\underset{\underline{\;}}{\tau }{}_{i}$, $\underset{\underline{\;}}{\phi }{}_{i}=T^{+}\underset{\underline{\;}}{\tau }{}_{i}-T^{-}{\bar{\tau }}_{i}$, ${\bar{\tau }}_{i}={\bar{B}}^{+}{{\bar{d}}_{i}}^{+}-{\underset{\underline{\;}}{B}}^{+}{{\bar{d}}_{i}}^{-}-{\bar{B}}^{-}{\underline{d}{}_{i}}^{+}+{\underset{\underline{\;}}{B}}^{-}{\underline{d}{}_{i}}^{-}$ and $\underset{\underline{\;}}{\tau }{}_{i}={\underset{\underline{\;}}{B}}^{+}{\underline{d}{}_{i}}^{+}-{\bar{B}}^{+}{\underset{\underline{\;}}{d}}^{-}-{\underset{\underline{\;}}{B}}^{-}{{\bar{d}}_{i}}^{+}+{\bar{B}}^{-}{{\bar{d}}_{i}}^{-}$. Moreover, from (6), we can deduce that
(7)$${\dot{\overset{⌢}{\;z}}}_{i}(t)=T(A(\theta ))-E(\theta )C)T^{-1}{\overset{⌢}{\;z}}_{i}(t)+T{\overset{⌢}{\;\tau }}_{i}$$
where ${\overset{⌢}{\;z}}_{i}={\bar{z}}_{i}-{\underline{z}}_{i}$ and ${\overset{⌢}{\tau }}_{i}={\bar{\tau }}_{i}-{\underline{\tau }}_{i}$.

**Lemma** **4**([41])**.**
*Under Assumptions 4 and 6, system (6) is an IO of system (5), i.e., if the initial states are set to ${\bar{z}}_{i}\left(0\right)=T^{+}{\bar{x}}_{i0}-T^{-}{\underset{\underline{\;}}{x}}_{i0}$ and ${\underset{\underline{\;}}{z}}_{i}\left(0\right)=T^{+}{\underset{\underline{\;}}{x}}_{i0}-T^{-}{\bar{x}}_{i0}$, then ${\underset{\underline{\;}}{z}}_{i}\left(t\right)\leq z_{i}\left(t\right)\leq {\bar{z}}_{i}\left(t\right)$ holds for all t≥0.*

**Lemma** **5**([38])**.**
*Under Assumptions 2, 3 and 6, and assuming that*
(1)*$∥{\bar{\tau }}_{i}∥<+\infty$ and $∥{\underset{\underline{\;}}{\tau }}_{i}∥<+\infty$ for any t≥0, ∥u∥≤U and all ${\bar{x}}_{i}\left(t\right)\in R^{n}$, ${\underset{\underline{\;}}{x}}_{i}\left(t\right)\in R^{n}$.*(2)*For any t⩾0, $∥x_{i}∥⩽{\bar{x}}_{i}$, $∥d_{i}∥⩽{\bar{d}}_{i}$ and $\bar{z}\left(t\right)\in R^{n}$, $\underline{z}\left(t\right)\in R^{n}$*
$${∥TB\left(\theta \right)d_{i}\left(t\right)-\underset{\underline{\;}}{\phi }{}_{i}∥}^{2}+{∥{\bar{\phi }}_{i}-TB\left(\theta \right)d_{i}\left(t\right)∥}^{2}\leq \chi_{i}{∥z_{i}-\underset{\underline{\;}}{z}{}_{i}∥}^{2}+\chi_{i}{∥{\bar{z}}_{i}-z_{i}∥}^{2}+\kappa_{i}$$
*hold for some $\chi_{i}\in R_{+}$, $\kappa_{i}\in R_{+}$ and $\chi_{i}I_{n}-TOT^{-1}+H\underset{-}{≺}$ 0, H = H*^T^ ≻ 0, *then the variables ${\bar{x}}_{i}(t)$ and
$\underline{x}{}_{i}(t)$ are bounded for all t* > 0 *and we have ${\underline{x}}_{i}(t)\leq x_{i}(t)\leq {\bar{x}}_{i}(t)$ provided that ${\underline{z}}_{i}(0)\leq z_{i}(0)\leq {\bar{z}}_{i}(0)$.*

**Proof****.** The upper and lower boundary estimation errors are defined as ${\bar{e}}_{z_{i}}={\bar{z}}_{i}-z_{i}$ and ${\underline{e}}_{z_{i}}=z_{i}-{\underline{z}}_{i}$, and from (5) and (6), we have
$$\left\{\begin{array}{c} {\dot{\bar{e}}}_{z_{i}}\left(t\right)=T\left(A\left(\theta \right)-E\left(\theta \right)C\right)T^{-1}{\bar{e}}_{z_{i}}\left(t\right)+{\bar{\phi }}_{i}-TB\left(\theta \right)d_{i}\left(t\right)\\ {\dot{\underline{e}}}_{z_{i}}\left(t\right)=T\left(A\left(\theta \right)-E\left(\theta \right)C\right)T^{-1}{\underline{e}}_{z_{i}}\left(t\right)+TB\left(\theta \right)d_{i}\left(t\right)-{\underline{\phi }}_{i} \end{array}\right.$$Given a matrix $E\left(\theta \right)$ of Assumption 2, for all t≥0 the properties ${\underline{\phi }}_{i}\leq TB\left(\theta \right)d_{i}\left(t\right)\leq {\bar{\phi }}_{i}$ and ${\underline{z}}_{i}\left(t\right)\leq z_{i}\left(t\right)\leq {\bar{z}}_{i}\left(t\right)$. In order to demonstrate that ${\bar{x}}_{i}\left(t\right)$ and ${\underline{x}}_{i}\left(t\right)$ are bounded, choosing the Lyapunov function $V_{i}={{\underset{\underline{\;}}{e}}_{z_{i}}}^{T}TF\left(t\right)T^{-1}{\underline{e}}_{z_{i}}+{{\bar{e}}_{z_{i}}}^{T}TF\left(t\right)T^{-1}{\bar{e}}_{z_{i}}$, its derivative is:
$$\begin{array}{c} \begin{array}{c} {\dot{V}}_{i}={{\underset{\underline{\;}}{e}}_{z_{i}}}^{T}T\left[\dot{F}\left(t\right)+N^{-1}\left(\theta \right)F\left(t\right)+F\left(t\right)N\left(\theta \right)\right]T^{-1}{\underset{\underline{\;}}{e}}_{z_{i}}+\\ {{\bar{e}}_{z}}^{T}T\left[\dot{F}\left(t\right)+N^{-1}\left(\theta \right)F\left(t\right)+F\left(t\right)N\left(\theta \right)\right]T^{-1}{\bar{e}}_{z_{i}}+\\ 2{{\underset{\underline{\;}}{e}}_{z_{i}}}^{T}TF\left(t\right)T^{-1}\left[TB\left(\theta \right)d_{i}\left(t\right)-{\underset{\underline{\;}}{\phi }}_{i}\right]+\\ 2{{\bar{e}}_{z_{i}}}^{T}TF\left(t\right)T^{-1}\left[{\bar{\phi }}_{i}-TB\left(\theta \right)d_{i}\left(t\right)\right] \end{array} \end{array}$$According to Assumption 6 this equality can take the following form:
$$\begin{array}{c} {\dot{V}}_{i}\leq -{{\underset{\underline{\;}}{e}}_{z_{i}}}^{T}TOT^{-1}{\underset{\underline{\;}}{e}}_{z_{i}}-{{\bar{e}}_{z_{i}}}^{T}TOT^{-1}{\bar{e}}_{z_{i}}+{\left|TB\left(\theta \right)d_{i}\left(t\right)-{\underset{\underline{\;}}{\phi }}_{i}\right|}^{2}+{\left|{\bar{\phi }}_{i}-TB\left(\theta \right)d_{i}\left(t\right)\right|}^{2} \end{array}$$If the first condition of the lemma is true, then the terms $∥TB\left(\theta \right)d_{i}\left(t\right)-{\underset{\underline{\;}}{\phi }}_{i}∥$ and $∥{\bar{\phi }}_{i}-TB\left(\theta \right)d_{i}\left(t\right)∥$ are bounded for any t≥0, $∥x_{i}∥\leq {\bar{x}}_{i}$, $∥d_{i}∥\leq {\bar{d}}_{i}$ and all ${\bar{z}}_{i}\left(t\right)\in R^{n}$, ${\underline{z}}_{i}\left(t\right)\in R^{n}$. Thus, the errors ${\underset{\underline{\;}}{e}}_{z_{i}}$ and ${\bar{e}}_{z_{i}}$ and the variables ${\bar{z}}_{i}\left(t\right)$ and ${\underline{z}}_{i}\left(t\right)$ are bounded. Because ${\bar{x}}_{i}\left(t\right)={\left(T^{-1}\right)}^{+}{\underline{z}}_{i}\left(t\right)-{\left(T^{-1}\right)}^{-1}{\bar{z}}_{i}\left(t\right)$ and ${\underline{x}}_{i}\left(t\right)={\left(T^{-1}\right)}^{+}{\bar{z}}_{i}\left(t\right)-{\left(T^{-1}\right)}^{-1}{\underline{z}}_{i}\left(t\right)$, the same conclusion can be drawn for the variables ${\bar{x}}_{i}\left(t\right)$ and ${\underline{x}}_{i}\left(t\right)$. Under the second condition of Lemma 5, we have:
$${\dot{V}}_{i}\leq -{{\underset{\underline{\;}}{e}}_{z_{i}}}^{T}H{\underset{\underline{\;}}{e}}_{z_{i}}-{{\bar{e}}_{z_{i}}}^{T}H{\bar{e}}_{z_{i}}+\kappa_{i}$$Notice that $y_{i}\left(t\right)=CT^{-1}\left(t\right)z_{i}\left(t\right)$, and based on Lemma 2, the interval estimation of $y_{i}\left(t\right)$ of the system is determined by the following equations
(8)$$\left\{\begin{array}{c} {\bar{y}}_{i}\left(t\right)={\left(CT^{-1}\right)}^{+}{\bar{z}}_{i}\left(t\right)-{\left(CT^{-1}\right)}^{-}{\underset{\underline{\;}}{z}}_{i}\left(t\right)\\ {\underset{\underline{\;}}{y}}_{i}\left(t\right)={\left(CT^{-1}\right)}^{+}{\underset{\underline{\;}}{z}}_{i}\left(t\right)-{\left(CT^{-1}\right)}^{-}{\bar{z}}_{i}\left(t\right) \end{array}\right.$$
such that ${\underline{y}}_{i}\left(t\right)\leq y_{i}\left(t\right)\leq {\bar{y}}_{i}\left(t\right)$ holds for all t≥0. □

**Assumption** **7.**
*The topology of leader–follower MASs (2) and (3) consists of an undirected spanning tree, with the leader agent as the root.*


**Lemma** **6.**
*Under Assumption 7, there exists a positive diagonal matrix G such that $GH+H^{T}G>0$. One such G is given by $G=diag\left(q_{1},\ldots ,q_{N}\right)$, where $q={\left[\begin{array}{ccc} q_{1} & \ldots & q_{N} \end{array}\right]}^{T}={\left(H^{T}\right)}^{-1}1_{N}$.*


**Lemma** **7.**
*For any nonzero $r\in R^{Nn}$, semi-positive-definite matrix I and symmetric matrix D, the following inequalities hold:*

$$\begin{array}{c} \lambda_{2}\left(D\right)r^{T}\left(I_{N}⊗I\right)r⩽r^{T}\left(D⊗I\right)r⩽\lambda_{\max}\left(D\right)r^{T}\left(I_{N}⊗I\right)r\\ \lambda_{2}\left(D\right)r^{T}\left(I⊗I_{N}\right)r⩽r^{T}\left(I⊗D\right)r⩽\lambda_{\max}\left(D\right)r^{T}\left(I⊗I_{N}\right)r \end{array}$$


*$\lambda_{2}\left(D\right)$ and $\lambda_{\max}\left(D\right)$ stand for the smallest non-zero and the maximum eigenvalues of D.*


### 3.2. UIO Design

In this subsection, an algebraic relationship between the UI and the state is first established based on the interval estimates determined by (8). Then, a Luenberger-like observer and the relationship constitute a UIO.

Denote $y_{i}={\left[\begin{array}{ccc} y_{i,1} & \cdots & y_{i,p} \end{array}\right]}^{T}$, ${\bar{y}}_{i}={\left[\begin{array}{ccc} {\bar{y}}_{i,1} & \cdots & {\bar{y}}_{i,p} \end{array}\right]}^{T}$ and ${\underline{y}}_{i}={\left[\begin{array}{ccc} {\underset{\underline{\;}}{y}}_{i,1} & \cdots & {\underset{\underline{\;}}{y}}_{i,p} \end{array}\right]}^{T}$. Since ${\underline{y}}_{i}⩽y_{i}⩽{\bar{y}}_{i}$ implies that ${\underline{y}}_{i,j}⩽y_{i,j}⩽{\bar{y}}_{i,j}$, we can conclude that there exist scalars $\beta_{i}\left(t\right)$ meeting with $0⩽\beta_{i,j}\left(t\right)⩽1$ so that $y_{i,j}=\beta_{i,j}\left({\bar{y}}_{i,j}-{\underset{\underline{\;}}{y}}_{i,j}\right)+{\underline{y}}_{i,j},\left(j=1,\cdots ,p\right)$. They can be written compactly as
(9)$$y_{i}=\mathrm{diag}\left({\overset{⌢}{\;y}}_{i}\right)\beta_{i}+{\underset{\underline{\;}}{y}}_{i}$$
where ${\overset{⌢}{\;y}}_{i}={\bar{y}}_{i}-{\underline{y}}_{i}$ and
${\beta_{i,1}=[\beta_{i,1}\cdots \beta_{i,j}]}^{T}$. In addition, it follows from (8) that ${\overset{⌢}{y}}_{i}=∥{CT}^{-1}(t)∥{\overset{⌢}{z}}_{i}$. Based on (7), it is very directly for one to obtain ${\dot{\overset{⌢}{\;y}}}_{i}=h_{i1}($${\overset{⌢}{z}}_{i}),\;\mathrm{where}$$$\begin{array}{c} h_{i1}\left({\overset{⌢}{\;z}}_{i}\right)=\left|CT^{-1}\left(t\right)\right|\left[T\left(A\left(\theta \right)-E\left(\theta \right)C\right)T^{-1}{\overset{⌢}{\;z}}_{i}\left(t\right)+T{\overset{⌢}{\;\tau }}_{i}\right] \end{array}$$

Using the second equation of (8) together with (6), we can deduce that $\dot{{\underset{\underline{\;}}{y}}_{i}}=h_{i2}({\bar{z}}_{i},{\underset{\underline{\;}}{z}}_{i},y_{i})+CB\left(\theta \right)u_{i}$, where
$$\begin{array}{c} h_{i,2}({\bar{z}}_{i},{\underset{\underline{\;}}{z}}_{i},y_{i})={\left(CT^{-1}\right)}^{+}T\left(A\left(\theta \right)-E\left(\theta \right)C\right)T^{-1}{\underset{\underline{\;}}{z}}_{i}-{\left(CT^{-1}\right)}^{-}T\left(A\left(\theta \right)-\right)\\ \left.E\left(\theta \right)C\right)T^{-1}{\bar{z}}_{i}+CLy_{i}+{\left(CT^{-1}\right)}^{+}{\underset{\underline{\;}}{\phi }}_{i}-{\left(CT^{-1}\right)}^{-}{\bar{\phi }}_{i} \end{array}$$

Consequently, it follows from (9) that
(10)$${\dot{y}}_{i}=\mathrm{diag}(h_{i1}({\overset{⌢}{\;z}}_{i}))\beta_{i}+\mathrm{diag}({\overset{⌢}{\;y}}_{i}){\dot{\beta }}_{i}+h_{i2}({\underset{\underline{\;}}{z}}_{i},{\bar{z}}_{i},y_{i})+CB\left(\theta \right)u_{i}$$

Moreover, from (3) and (10), we can infer that
(11)$$CB\left(\theta \right)d_{i}=\mathrm{diag}(h_{i1}({\overset{⌢}{\;z}}_{i}))\beta_{i}+\mathrm{diag}({\overset{⌢}{\;y}}_{i}){\dot{\beta }}_{i}+h_{i2}({\underset{\underline{\;}}{z}}_{i},{\bar{z}}_{i},y_{i})-CA\left(\theta \right)x_{i}$$

Under Assumption 5, we can obtain from (11) that
(12)$$d_{i}={\left(CB\left(\theta \right)\right)}^{†}[\mathrm{diag}(h_{i1}({\overset{⌢}{\;z}}_{i}))\beta_{i}+\mathrm{diag}({\overset{⌢}{\;y}}_{i}){\dot{\beta }}_{i}+h_{i2}({\underset{\underline{\;}}{z}}_{i},{\bar{z}}_{i},y_{i})-CA\left(\theta \right)x_{i}]$$

Moreover, it follows from (9) that
(13)$$\beta_{i}=\left\{{\left[\mathrm{diag}({\overset{⌢}{\;y}}_{i}+\eta_{i})\right]}^{-1}-\mathrm{diag}\left(\eta_{i}\right)\right\}\left(y_{i}-{\underset{\underline{\;}}{y}}_{i}\right)$$
where $\eta_{i}={\left[\begin{array}{ccc} \eta_{i,1} & \cdots & \eta_{i,p} \end{array}\right]}^{T}$ with $\eta_{i,j}=1$ if ${\bar{y}}_{i,j}={\underline{y}}_{i,j}$; otherwise, $\eta_{i,j}=0,\left(j=1,\cdots ,p\right)$. Now, referring to (3) and (12), a UIO is designed for each follower agent of system (3) as shown below:
(14)$$\left\{\begin{array}{c} {\dot{\hat{x}}}_{i}=A\left(\theta \right){\hat{x}}_{i}+B\left(\theta \right)u_{i}+B\left(\theta \right){\hat{d}}_{i}+L\left(\theta \right)\left(C{\hat{x}}_{i}-y_{i}\right)\\ {\hat{d}}_{i}={\left(CB\left(\theta \right)\right)}^{†}\left[\mathrm{diag}(h_{i1}({\overset{⌢}{\;z}}_{i}))\beta_{i}+\mathrm{diag}({\overset{⌢}{\;y}}_{i})\hat{\dot{\beta_{i}}}+h_{i2}(z_{i},{\bar{z}}_{i},y_{i})-CA(\theta ){\hat{x}}_{i}\right] \end{array}\right.$$
where $\beta_{i}$ is determined by (13) and $\hat{\dot{\beta_{i}}}$ is generated by the differentiator (1), which is the same estimate of ${\dot{\beta }}_{i}$ at a finite time frame.

**Theorem** **1.**
*Under Assumptions 4, 5 and 6, system (14) can simultaneously produce asymptotic UIRs denoted by ${\hat{d}}_{i}$ and asymptotic state estimates denoted by ${\hat{x}}_{i}$. Moreover, (14) is a UIO of system (3). Define $e_{i}\left(t\right)={\hat{x}}_{i}\left(t\right)-x_{i}\left(t\right)$ and ${\tilde{d}}_{i}\left(t\right)={\hat{d}}_{i}\left(t\right)-d_{i}\left(t\right)$. Then both $\lim_{t\to \infty }e_{i}\left(t\right)=0$ and $\lim_{t\to \infty }{\tilde{d}}_{i}\left(t\right)=0$ hold.*


**Proof****.** The system of observer error dynamics is available from (3) and the first equation of (14):
(15)$${\dot{e}}_{i}\left(t\right)=\left[A\left(\theta \right)-L\left(\theta \right)C\right]e_{i}\left(t\right)+B\left(\theta \right){\tilde{d}}_{i}\left(t\right)$$Since the estimation of $\hat{\dot{\beta_{i}}}$ is derived from the differentiator (1), it is able to estimate ${\dot{\beta }}_{i}$ identically in a finite time. Defining $\tilde{\dot{\beta_{i}}}\left(t\right)=\hat{\dot{\beta_{i}}}\left(t\right)-{\dot{\beta }}_{i}\left(t\right)$, we can obtain $\tilde{\dot{\beta_{i}}}\left(t\right)=0$ in a finite time. Then, combining (12) and the second equation of (14), we obtain
(16)$${\tilde{d}}_{i}\left(t\right)=-{\left(CB\left(\theta \right)\right)}^{†}CA\left(\theta \right)e_{i}\left(t\right)$$Moreover, substituting (16) into (15) yields
(17)$${\dot{e}}_{i}\left(t\right)=\left\{\left[I_{n}-B\left(\theta \right){\left(CB\left(\theta \right)\right)}^{†}C\right]A\left(\theta \right)-L\left(\theta \right)C\right\}e_{i}\left(t\right)$$Then, based on Theorem 2, choosing the appropriate observer gain matrix $L\left(\theta \right)$ makes (17) an asymptotic dynamic system. Therefore, $\lim_{t\to \infty }e_{i}\left(t\right)=0$. Now, bringing back (16), we can obtain $\lim_{t\to \infty }{\tilde{d}}_{i}\left(t\right)=0$. □

**Remark** **1.**
*For the follower systems (3), through designing an interval observer given by (6), an algebraic relationship has been established by the UI and system states, and it is given by (12). By combining the algebraic relationship with a Luenberger-like observer, a UIO is constructed, which is described by (14). The design of the UIO allows simultaneous asymptotic estimation of the follower’s state and reconstruction of the unknown input. In addition, another significant feature of the UIR is that it successfully decouples the control inputs. This characteristic will facilitate the design of a compensation controller to compensate for the unknown input.*


## 4. UIO-Based Distributed Consensus

In this section, a distributed control protocol scheme is developed based on the state estimation and unknown input reconstruction provided by (14). Under the distributed control protocol scheme, the leader–follower consensus of the LPV MAS by Definition 2 is accomplished.

Consider the following distributed control protocol constructed using UIR and the state estimation:
(18)$$u_{i}\left(t\right)=K\left(\theta \left(t\right)\right)\left[\sum_{j=}^{N}a_{ij}\left({\hat{x}}_{i}\left(t\right)-{\hat{x}}_{j}\left(t\right)\right)+b_{i}\left({\hat{x}}_{i}\left(t\right)-x_{0}\left(t\right)\right)\right]-{\hat{d}}_{i}\left(t\right),i=1,\ldots ,N$$

The estimated states ${\hat{x}}_{i}\left(t\right)$ and UIR ${\hat{d}}_{i}\left(t\right)$ are provided by local UIO (14). Introduce the notation $\Phi \left(\theta \right)=B\left(\theta \right){\left(CB\left(\theta \right)\right)}^{†}$ and suppose it can also be described by $\Phi \left(\theta \right)=\sum_{h=1}^{S}\rho_{h}\left(\theta \right)\Phi_{h}$. Furthermore, denote $I_{n}-\Phi \left(\theta \right)C$; obviously, we have $H\left(\theta \right)=\sum_{l=1}^{S}\rho_{l}\left(\theta \right)H_{h}$ with $I_{n}-\Phi_{l}C_{l}$. Moreover, let $L\left(\theta \right)=\sum_{h=1}^{S}\rho_{i}\left(\theta \right)L_{h}$ and $K\left(\theta \right)=\sum_{h=1}^{S}\rho_{i}\left(\theta \right)K_{h}$ be, respectively, the observer and control gain matrices to be designed later and define
(19)$$\varpi_{lh}=P_{1}A_{h}+A_{h}^{T}P_{1}-\frac{\lambda_{2}\left(M\right)}{\lambda_{\max}\left(G\right)}P_{1}B_{l}B_{l}^{T}P_{1}$$
(20)$$\Omega_{lh}=\left[\begin{array}{cc} G⊗\varpi_{lh} & -\left(GH\right)⊗\left(P_{1}B_{l}B_{l}^{T}P_{1}\right)-G⊗\left(P_{1}\Phi_{l}A_{h}\right)\\ {}* & I_{N}⊗He\left\{P_{2}H_{h}A_{l}-X_{h}C_{l}\right\} \end{array}\right]$$
Then we have the following result.

**Theorem** **2.**
*Suppose that the following matrix inequalities*

(21)
$$\Omega_{lh}≺0,\;\left(l,h=1,\cdots ,S\right)$$

*are feasible for $P_{1}≻0$, $P_{2}≻0$ and $X_{h}\left(h=1,\cdots ,S\right)$, and choose $K_{h}=-B_{h}^{T}P_{1}$ and $L_{h}=P_{2}^{-1}X_{h}$. Then the UIO-based distributed controller (19) can realize the consensus of MAS (2) and (3) in the sense of Definition 2.*


**Proof****.** First, by (2), (3) and (18), we have
(22)$$\begin{array}{c} {\dot{\delta }}_{i}=A\left(\theta \right)\delta_{i}+B\left(\theta \right)K\left(\theta \right)\left[\sum_{j=1}^{N}a_{ij}\left({\hat{x}}_{i}-{\hat{x}}_{j}\right)+b_{i}\left({\hat{x}}_{i}-x_{0}\right)\right]-B\left(\theta \right){\tilde{d}}_{i}\\ =A\left(\theta \right)\delta_{i}+B\left(\theta \right)K\left(\theta \right)\sum_{j=1}^{N}a_{ij}\left(e_{i}-e_{j}\right)+B\left(\theta \right)K\left(\theta \right)b_{i}e_{i}\\ +B\left(\theta \right)K\left(\theta \right)\sum_{j=1}^{N}a_{ij}\left(\delta_{i}-\delta_{j}\right)+B\left(\theta \right)K\left(\theta \right)b_{i}\delta_{i}-B\left(\theta \right){\tilde{d}}_{i} \end{array}$$Substituting (16) into (22) gives
(23)$$\begin{array}{c} {\dot{\delta }}_{i}=A\left(\theta \right)\delta_{i}+B\left(\theta \right)K\left(\theta \right)\sum_{j=1}^{N}a_{ij}\left(e_{i}-e_{j}\right)+B\left(\theta \right)K\left(\theta \right)b_{i}e_{i}\\ +B\left(\theta \right)K\left(\theta \right)\sum_{j=1}^{N}a_{ij}\left(\delta_{i}-\delta_{j}\right)+B\left(\theta \right)K\left(\theta \right)b_{i}\delta_{i}-B\left(\theta \right){\left(CB\left(\theta \right)\right)}^{†}CA\left(\theta \right)e_{i}\left(t\right) \end{array}$$The overall system of (23) is
(24)$$\begin{array}{c} \dot{\delta }=\left[I_{N}⊗A\left(\theta \right)+H⊗\left(B\left(\theta \right)K\left(\theta \right)\right)\right]\delta +\left[H⊗\left(B\left(\theta \right)K\left(\theta \right)\right)-I_{N}⊗\left(\Phi \left(\theta \right)CA\left(\theta \right)\right)\right]e \end{array}$$
where $\Phi \left(\theta \right)=B\left(\theta \right){\left(CB\left(\theta \right)\right)}^{†}$, $\delta ={\left[\begin{array}{ccc} \delta_{1}^{T} & \cdots & \delta_{N}^{T} \end{array}\right]}^{T}$ and ${\left[\begin{array}{ccc} e_{1}^{T} & \cdots & e_{N}^{T} \end{array}\right]}^{T}$. On the other hand, the overall system of (17) is
(25)$$\dot{e}=\left[I_{N}⊗\left(H\left(\theta \right)A\left(\theta \right)-L\left(\theta \right)C\right)\right]e$$Now, combining (24) and (25), we have
(26)$$\left[\begin{array}{c} \dot{\delta }\\ \dot{e} \end{array}\right]=\left[\begin{array}{cc} I_{N}⊗A\left(\theta \right)+H⊗\left(B\left(\theta \right)K\left(\theta \right)\right) & H⊗\left(B\left(\theta \right)K\left(\theta \right)\right)-I_{N}⊗\left(\Phi \left(\theta \right)A\left(\theta \right)\right)\\ 0 & I_{N}⊗\left(H\left(\theta \right)A\left(\theta \right)-L\left(\theta \right)C\right) \end{array}\right]\left[\begin{array}{c} \delta \\ e \end{array}\right]$$Denote $z={\left[\begin{array}{cc} \delta^{T} & e^{T} \end{array}\right]}^{T}$ and consider the Lyapunov function
$$V\left(z\right)=V_{1}\left(\delta \right)+V_{2}\left(e\right)$$
with $V_{1}\left(\delta \right)=\delta^{T}\left(G⊗P_{1}\right)\delta$ and $V_{2}\left(e\right)=e^{T}\left(I_{N}⊗P_{2}\right)e$. Set $K\left(\theta \right)=-B^{T}\left(\theta \right)P_{1}$ and recall that $M=GH+H^{T}G$ as defined in Lemma 6 and Lemma 7. Then
$$\begin{array}{c} {\dot{V}}_{1}\left(\delta \right)=\delta^{T}\left[G⊗\left(P_{1}A\left(\theta \right)+A\left(\theta \right)P_{1}\right)-M⊗\left(P_{1}B\left(\theta \right)B^{T}\left(\theta \right)P_{1}\right)\right]\delta -\\ 2\delta^{T}\left[GH⊗\left(P_{1}B\left(\theta \right)B^{T}\left(\theta \right)P_{1}\right)+G⊗\left(P_{1}\Phi \left(\theta \right)A\left(\theta \right)\right)\right]e\\ \leq \delta^{T}\left[G⊗\left(P_{1}A\left(\theta \right)+A\left(\theta \right)P_{1}\right)-\frac{\lambda_{2}\left(M\right)}{\lambda_{\max}\left(G\right)}G⊗\left(P_{1}B\left(\theta \right)B^{T}\left(\theta \right)P_{1}\right)\right]\delta -\\ 2\delta^{T}\left[GH⊗\left(P_{1}B\left(\theta \right)B^{T}\left(\theta \right)P_{1}\right)+G⊗\left(P_{1}\Phi \left(\theta \right)A\left(\theta \right)\right)\right]e \end{array}$$
and
$${\dot{V}}_{2}\left(e\right)=e^{T}\left\{I_{N}⊗\left[P_{2}\left(H\left(\theta \right)A\left(\theta \right)-L\left(\theta \right)C\right)+{\left(H\left(\theta \right)A\left(\theta \right)-L\left(\theta \right)C\right)}^{T}P_{2}\right]\right\}e$$That is, we have
$$\dot{V}\left(z\right)\leq z^{T}\Omega \left(\theta \right)z$$
where
(27)$$\begin{array}{c} \Omega \left(\theta \right)=\left[\begin{array}{c} G⊗\left[\left(P_{1}A\left(\theta \right)+A^{T}\left(\theta \right)P_{1}\right)-\frac{\lambda_{2}\left(M\right)}{\lambda_{\max}\left(G\right)}P_{1}B\left(\theta \right)B^{T}\left(\theta \right)P_{1}\right]\\ {}* \end{array}\right.\\ \left.\;\;\;\;\;\;\;\;\;\;\;\;\;\;\;\;\;\;\;\;\;\;\;\;\;\;\begin{array}{c} -GH⊗\left(P_{1}B\left(\theta \right)B^{T}\left(\theta \right)P_{1}\right)-G⊗\left(P_{1}\Phi \left(\theta \right)A\left(\theta \right)\right)\\ I_{N}⊗P_{2}\left(H\left(\theta \right)A\left(\theta \right)-L\left(\theta \right)C\right)+{\left(H\left(\theta \right)A\left(\theta \right)-L\left(\theta \right)C\right)}^{T}P_{2} \end{array}\right] \end{array}$$Recall that $\Phi \left(\theta \right)=\sum_{l=1}^{S}\rho_{l}\left(\theta \right)\Phi_{l}$, $A\left(\theta \right)=\sum_{l=1}^{S}\rho_{l}\left(\theta \right)A_{l}$, $B\left(\theta \right)=\sum_{l=1}^{S}\rho_{l}\left(\theta \right)B_{l}$, $H\left(\theta \right)=\sum_{l=1}^{S}\rho_{l}\left(\theta \right)H_{h}$, $L\left(\theta \right)=\sum_{l=1}^{S}\rho_{l}\left(\theta \right)L_{l}$ and $K\left(\theta \right)=\sum_{l=1}^{S}\rho_{l}\left(\theta \right)K_{l}$. Then we have $\Omega \left(\theta \right)=\sum_{l=1}^{S}\sum_{h=1}^{S}\rho_{l}\left(\theta \right)\rho_{h}\left(\theta \right)\Omega_{lh}$, where $\Omega_{lh}$ is defined by (20). Therefore, (21) and (27) imply that $\dot{V}\left(z\right)<0$, which means the closed-loop system (26) is asymptotically stable.In the following, we offer the Algorithm 1 of calculating the observer gain matrix and control matrix. □

**Algorithm 1** Obatin observer gain and controller gain.**Step 1:** For a proper symmetric positive matrix Q≻0, solve the Riccati equation $\varpi_{lh}+Q=0$ to obtain $P_{1}≻0$;**Step 2:** Solve the LMIs (22) in **Theorem 2** and obtain $P_{2}≻0$ and matrix $X_{h}\left(h=1,\cdots ,N\right)$;**Step 3:** By $K_{h}=-B_{h}^{T}P_{1}$ and $L_{h}=P_{2}^{-1}X_{h}$, obtain $K_{h}$ and $L_{h}\left(h=1,\cdots ,N\right)$.**Step 4:** The observer gain and the controller gain matrices can be determined by $L\left(\theta \right)=\sum_{l=1}^{S}\rho_{l}\left(\theta \right)L_{l}$ and $K\left(\theta \right)=\sum_{l=1}^{S}\rho_{l}\left(\theta \right)K_{l}$, respectively.

## 5. Simulation Analysis

In this section, a simulation example is given to verify the effectiveness of the proposed method.

Consider an LPV MAS. The leader is in form (2) and the follower agent is in form (3) with
$$\begin{array}{c} A\left(\theta \left(t\right)\right)=\left[\begin{array}{cc} -0.632 & \theta \left(t\right)\\ -\theta \left(t\right) & 0 \end{array}\right],B\left(\theta \left(t\right)\right)=\left[\begin{array}{c} \theta \left(t\right)\\ \theta \left(t\right) \end{array}\right],C=\left[\begin{array}{cc} 0 & 1 \end{array}\right] \end{array}$$

Moreover, by the definitions of $\Phi \left(\theta \right)$ and $H\left(\theta \right)=I_{n}-\Phi \left(\theta \right)C$, we can calculate that $\Phi \left(\theta \left(t\right)\right)=\left[\begin{array}{cc} 0 & 1\\ 0 & 1 \end{array}\right]$. The parameter $\theta \left(t\right)$ is supposed as $\theta \left(t\right)=0.5+0.1\sin\left(t\right)$, which obviously satisfies $0.4\leq \theta \left(t\right)\leq 0.6$. Moreover, choosing $\rho_{1}\left(t\right)=\frac{0.6-\theta \left(t\right)}{0.2}$ and $\rho_{2}\left(t\right)=\frac{\theta \left(t\right)-0.4}{0.2}$, we have
$$\begin{array}{c} \begin{array}{c} A_{1}=\left[\begin{array}{cc} -0.632 & 0.4\\ -0.4 & 0 \end{array}\right],A_{2}=\left[\begin{array}{cc} -0.632 & 0.6\\ -0.6 & 0 \end{array}\right],B_{1}=\left[\begin{array}{c} 0.4\\ 0.4 \end{array}\right],B_{2}=\left[\begin{array}{c} 0.6\\ 0.6 \end{array}\right]\\ \Phi_{1}=\left[\begin{array}{cc} 0 & 1\\ 0 & 1 \end{array}\right],\Phi_{2}=\left[\begin{array}{cc} 0 & 1\\ 0 & 1 \end{array}\right],H_{1}=\left[\begin{array}{cc} 1 & -1\\ 0 & 0 \end{array}\right],H_{2}=\left[\begin{array}{cc} 1 & -1\\ 0 & 0 \end{array}\right] \end{array} \end{array}$$

Disturbances of each agent and their upper and lower bounds are set as
$$\begin{array}{c} \begin{array}{c} d_{1}\left(t\right)=\sin\left(2t\right)+1.5\cos\left(3t\right),{\underline{d}}_{1}=-3,{\bar{d}}_{1}=3;\\ d_{2}\left(t\right)=2sawtooth\left(\frac{1}{2}\pi t+\frac{1}{2}\pi \right),{\underline{d}}_{2}=-3,{\bar{d}}_{2}=3;\\ d_{3}\left(t\right)=1.5sawtooth\left(\pi t+\frac{1}{3}\pi ,0\right),{\underline{d}}_{3}=-2,{\bar{d}}_{3}=2;\\ d_{4}\left(t\right)=2.5sawtooth\left(\frac{1}{3}\pi t+\frac{1}{2}\pi ,\frac{1}{2}\right),{\underline{d}}_{4}=-5,{\bar{d}}_{4}=3; \end{array} \end{array}$$

According to the communication graph of Figure 1, the Laplacian matrix L and a matrix B can be obtained.
$$\begin{array}{c} L=\left[\begin{array}{cccc} 1 & 0 & 0 & -1\\ -1 & 1 & 0 & 0\\ 0 & -1 & 1 & 0\\ 0 & -1 & -1 & 2 \end{array}\right],B=\left[\begin{array}{cccc} 1 & 0 & 0 & 0\\ 0 & 0 & 0 & 0\\ 0 & 0 & 0 & 0\\ 0 & 0 & 0 & 0 \end{array}\right] \end{array}$$

According to Theory 2, we solve LMIs (23) and obtain
$$\begin{array}{c} P_{1}=\left[\begin{array}{cc} 433.8717 & -49.2137\\ -49.2137 & 210.0338 \end{array}\right],P_{2}=\left[\begin{array}{cc} 1.5023 & -1.1275\\ -1.1275 & 3.2402 \end{array}\right] \end{array}$$

The controller gains are
$$\begin{array}{c} \begin{array}{c} K_{1}=\left[\begin{array}{cc} -0.1499 & -0.8451 \end{array}\right],K_{2}=\left[\begin{array}{cc} -0.2249 & -1.2677 \end{array}\right]\\ \sum_{h=1}^{2}\alpha_{h}\left(t\right)K_{h}=\left[\begin{array}{cc} 1\times 10^{-4}-0.375\times \theta \left(t\right) & 1\times 10^{-4}-2.113\times \theta \left(t\right) \end{array}\right] \end{array} \end{array}$$

The observer gains are
$$\begin{array}{c} L_{1}=\left[\begin{array}{c} -0.6479\\ -0.5527 \end{array}\right],L_{1}=\left[\begin{array}{c} -0.6353\\ -0.5641 \end{array}\right],\sum_{h=1}^{2}\alpha_{h}\left(t\right)L_{h}=\left[\begin{array}{c} -0.6731+0.063\times \theta \left(t\right)\\ -0.5281-0.06\times \theta \left(t\right) \end{array}\right] \end{array}$$

We obtain the following simulation results: The results of the simulation are presented in Figure 2, Figure 3, Figure 4 and Figure 5. Figure 2 and Figure 3 show the state $x_{i}$ and the estimation of the state ${\hat{x}}_{i}$ by the first equation of UIO (14), showing that the state estimation converges to the state within a finite time. Figure 4 gives the unknown input reconstruction result, proving that the second equation of (14) can realize the reconstruction of the unknown input $d_{i}$ successfully. The simulation results in Figure 5 indicate that under the distributed UIO-based controller (18), the follower agent state $x_{i}$ can synchronize the leader agent state $x_{0}$ within a finite time.

Next, we make a comparison to the work of [32]. In [32], for LPV MASs without unknown disturbances, an observer-based distributed controller whose gain has variable parameters was designed. Figure 6 offers the consensus effects when every agent system does not suffer from unknown disturbance (Figure 6a) and suffers from unknown disturbance (Figure 6b) under the observer-based distributed controller given by [32] (due to space limitations, only the consensus for the state $x_{1}$ is provided). For the method proposed in [32], from Figure 6a, we find that the consensus can be reached if the system suffers from no disturbance. Figure 6b shows that the method proposed in [32] cannot work well if the system has disturbance. In our method, however, because of the development of the UIO, which can offer the state estimation and unknown input reconstruction asymptotically, and because of the introduction of the UIR into the distributed controller, the asymptotic convergence consensus can still be accomplished even if every agent in the MAS suffers from unknown disturbance. Therefore, we can conclude that our method has the advantage that the distributed controller has a strong ability to deal with unknown inputs. One of the disadvantages of our method is that its structure is relatively complex, requiring the design of an interval observer for each follower agent.

## 6. Conclusions

This paper has studied the distributed consensus control problems of linear parameter-varying multi-agent systems with unknown inputs. To begin with, for each LPV agent, an interval observer is designed, and an algebraic relationship between the UI and the state is put forward. After this, a UIR method is developed by referring to the algebraic relationship, and then a UIO is designed by combining a Luenberger-like observer with the UIR. Moreover, a UIO-based distributed control protocol scheme is developed. Our methods show some advantages. Firstly, the proposed UIO can offer the asymptotic convergence state estimation and unknown input reconstruction simultaneously. Secondly, the unknown input reconstruction decouples the control input successfully. Thirdly, the distributed controller can realize asymptotic convergence consensus of an MAS even if the MAS suffers from external disturbances. Further research will focus on improving the proposed method to solve the problems of the bipartite consensus and containment consensus.

## Figures and Tables

**Figure 1 sensors-23-05125-f001:**
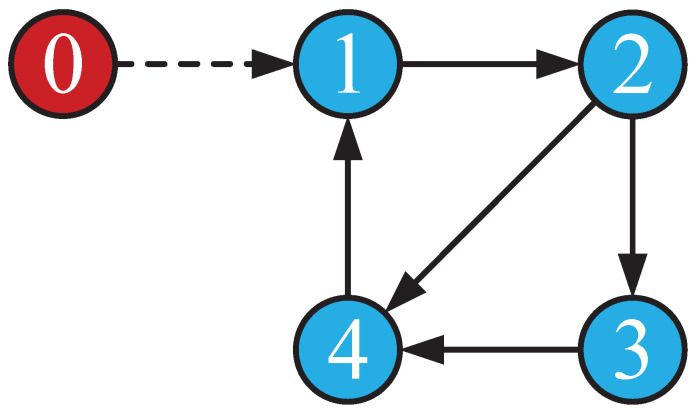
Communication graph.

**Figure 2 sensors-23-05125-f002:**
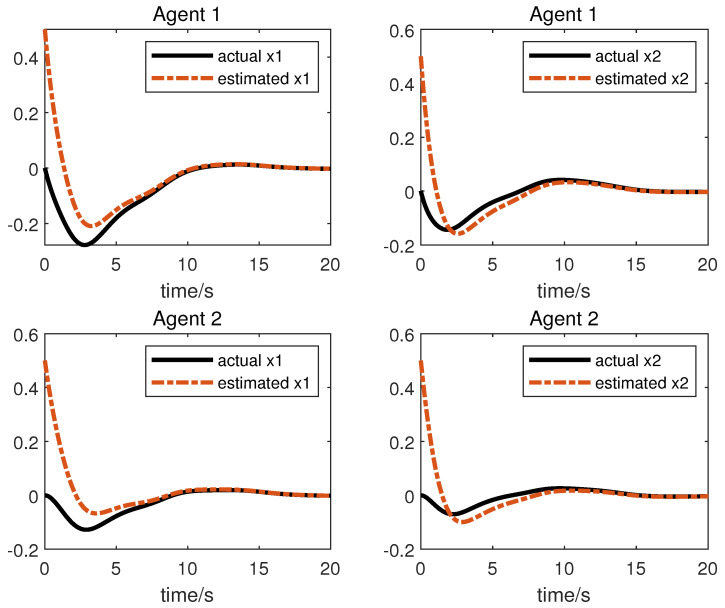
State $x_{i}$ and its estimations ${\hat{x}}_{i}$.

**Figure 3 sensors-23-05125-f003:**
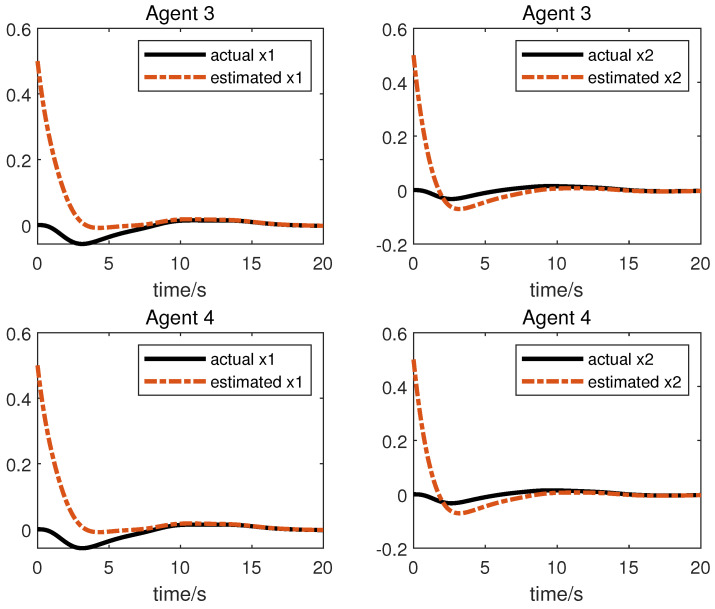
State $x_{i}$ and its estimations ${\hat{x}}_{i}$.

**Figure 4 sensors-23-05125-f004:**
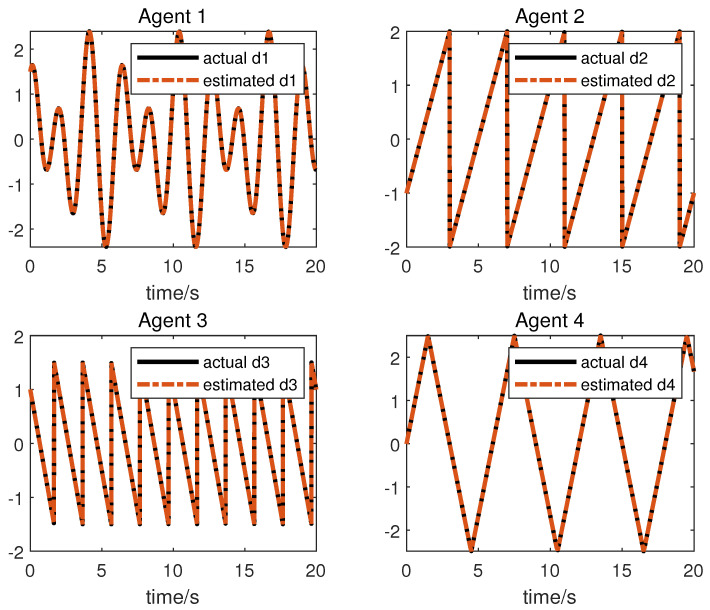
The unknown input $d_{i}$ and its reconstruction $\hat{d_{i}}$.

**Figure 5 sensors-23-05125-f005:**
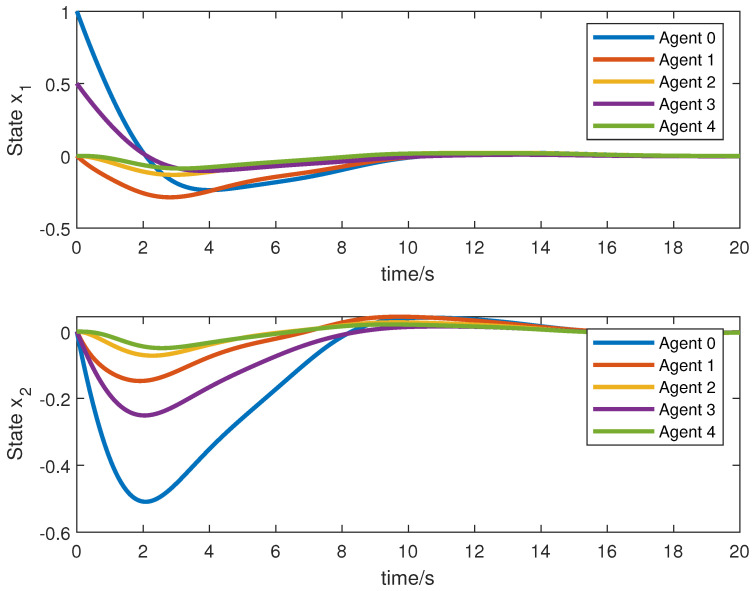
Trajectories of the leader and the follower agents’ states under (18).

**Figure 6 sensors-23-05125-f006:**
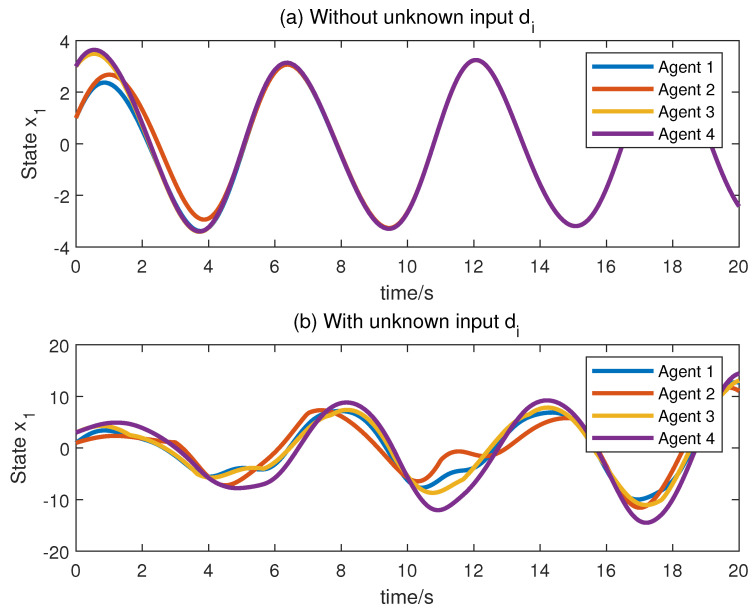
The consensus of state $x_{1}$ under the distributed controller given by [32].

## Data Availability

No new data were created.

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
