# Peer review of "Consensus Control of Linear Parameter-Varying Multi-Agent Systems with Unknown Inputs"

_sensors, 2023, doi:10.3390/s23115125_

Round 1
Reviewer 1 Report
This paper provided the consensus control problems of linear parameter-varying multi-agent systems with unknown input. Generally, this is quite interesting work. It can be accepted if the authors can consider the following issues:
The main reasons for such a recommendation are listed below:
1. The literature review needs to improve by the latest published works as follows:
A) A Novel Event-Triggered Consensus Approach for Generic Linear Multi-Agents Under Heterogeneous Sector-Restricted Input Nonlinearities
B) Consensus tracking of multi-agent systems using constrained neural-optimizer-based sliding mode control
C) 2. Equations 2 and 3, have no citations.
3. After equation (3), how is the disturbance assumed? How is the disturbance boundary defined?
4. In the simulation results section, how to verify this example?
5. Figures 2-5 do not have enough resolution and should be replotted. Increase the width of the lines and use dark colors like black in the design.
6. The simulation Figures lack the units at their axes
7. For the simulation results, more explanations on them seem necessary and helpful to readers.
8. It is better to compare with other methods to give the readers an idea of ​​how your proposed method looks-wise.
9. Is the proposed method generalizable for other problems? Need more discussion on this issue.
10. Can experiments be done to show the advantage of the proposed method, for example, in robots?
11. The conclusion should clearly state the technique's main contributions and novelty.
In conclusion, I believe this manuscript is worthy of publication in the Journal of Sensors, after a minor revision.
There are some typo and grammatical errors which can be corrected.
Reviewer 2 Report
An asymptotical consensus control protocol scheme based on the unknown input observer is put forward in this manuscript. Based on the state estimation and the UIR, the distributed control protocol scheme is developed such that the asymptotic convergent consensus can be accomplished. After reading the entire article, I think it has the following shortcomings.
1. In the experimental part, the effectiveness of the method is verified using sinusoidal signals as input. To make the experimental results more convincing, the experiment should select more representative signals as input or explain the principle of input signal selection and why only sinusoidal signals were chosen as unknown signals.
2. Please check and supplement the number of the formula in the text.
3. In the second section, formula derivation and theoretical analysis are carried out, where symbols are mixed with Latin and English symbols. The paper does not indicate whether different symbols have different meanings, and it also mixes different styles of symbols in the same formula, such as in Assumption 2. The significance of the author's doing this or the change of characters due to the typesetting is unclear. The typesetting version should be checked, or the author should supplement the relevant operations. For example, the term "Matrix greater than zero" should be explained to mean that the matrix is positive definite as the reference [36].
4. It would be helpful if the authors could uniformly list abbreviations such as IO (Interval Observer) for easy reading.
5. The author can appropriately increase the comparative analysis with other studies in this direction and discuss the advantages and disadvantages of this method
6. When verifying the proposed observer and controller design methods, or selecting more types of signal input to prove the universality of the method, or selecting a class of signals for targeted research. In any case, the basis and reasons for the choice should be pointed out. In general, the advantages, application scope and shortcomings of the method are pointed out in the paper, and the innovation points of the research are highlighted. The various operations included in the algorithm should be specified, rather than a reference to the relevant literature.
The English language should be thoroughly polished and checked. All typos should be corrected, and the entire manuscript should be proofread and edited. For example, pay attention to the use of definite articles and indefinite articles. And pay attention to the spelling of words, such as the first sentence in section 5.
Round 2
Reviewer 1 Report
I have no further comments.
I have no further comments.
Reviewer 2 Report
All my concerned problems have been properly considered in the revised manuscript, so I suggest its publication in Sensors.